

# The impact of leopards (*Panthera pardus*) on livestock losses and human injuries in a human-use landscape in Maharashtra, India

Vidya Athreya[1], Kavita Isvaran[2], Morten Odden[3], John D.C. Linnell[4], Aritra Kshettry[1,5,6], Jagdish Krishnaswamy[7] and Ullas K. Karanth[6,8]

[1] Wildlife Conservation Society-India, Bengaluru, Karnataka, India
[2] Centre for Ecological Sciences, Indian Institute of Science, Bengaluru, Karnataka, India
[3] Faculty of Applied Ecology, Agricultural Sciences and Biotechnology, Inland Norway University of Applied Sciences, Evenstad, Koppang, Norway
[4] Norwegian Institute of Nature Research, Trondheim, Norway
[5] INSPIRE-Fellow, Department of Science and Technology, New Delhi, India
[6] Centre for Wildlife Studies, Bengaluru, Karnataka, India
[7] Ashoka Trust for Research in Ecology and the Environment, Bengaluru, Karnataka, India
[8] Wildlife Conservation Society, New York, NY, United States of America

Corresponding author
Vidya Athreya, vidya@wcsindia.org, vidya.athreya@gmail.com

## ABSTRACT

There are many ways in which large carnivores and humans interact in shared spaces. In this study we provide insights into human-leopard relationships in an entirely modified, human-dominated landscape inhabited by dense populations of humans (266 per km$^2$), their livestock (162 per km$^2$) and relatively high densities of large predators (10 per 100 km$^2$). No human deaths were recorded, and livestock losses to leopards numbered only 0.45 per km$^2$ per year (averaged over three years) despite the almost complete dependency of leopards on domestic animals as prey. Predation was not the major cause of livestock mortality as diseases and natural causes caused higher losses (80% of self-reported losses). We also found that ineffective night time livestock protection and the presence of domestic dogs increased the probability of a farmer facing leopard attacks on livestock. Resident farmers faced much lower livestock losses to leopard predation in contrast to the migratory shepherds who reported much higher losses, but rarely availed of the government compensation schemes. We recommend that local wildlife managers continue to shift from reactive measures such as leopard captures after livestock attacks to proactive measures such as focusing on effective livestock protection and informing the affected communities about safety measures to be taken where leopards occur in rural landscapes. The natural causes of livestock deaths due do diseases may be better prevented by involving animal husbandry department for timely vaccinations and treatment.

## INTRODUCTION

Conflicts between humans and large felids have been at the centre of attention in conservation because of the impact these threatened species have on the lives and livelihoods of local people (*Treves & Karanth, 2003*; *Inskip & Zimmermann, 2009*; *Ripple et al., 2014*), as well as the decline of wild felid populations which can occur due to retaliatory killings (*Inskip et al., 2014*). Livestock depredation, resulting in real or perceived economic losses to individual farmers, is the most common cause of human-large felid conflict (*Ogada et al., 2003*; *Katel, Pradhan & Schmidt-Vogt, 2014*; *Peña Mondragón et al., 2017*; *Suryawanshi et al., 2017*) although some species of large felids are also associated with attacks on humans (*Athreya et al., 2011*; *Kshettry, Vaidyanathan & Athreya, 2017*; *Packer et al., 2019*). The prevention and mitigation of conflicts is a challenging issue not only because of its urgency as many large felids are threatened, but solutions have to take into account complex and locally dependent social and cultural aspects (*Barlow et al., 2010*; *Dickman, 2010*; *Redpath et al., 2013*).

There is increasing evidence that poor livestock protection practices, sometimes in combination with low wild prey density, lead to livestock depredation by large felids (*Athreya et al., 2016*; *Shehzad et al., 2015*; *Suryawanshi et al., 2017*; *Kshettry, Vaidyanathan & Athreya, 2018*). Most of the information on livestock depredation by large felids in tropical areas are from in, or around, protected areas (*Patterson et al., 2004*; *Harihar, Pandav & Goyal, 2011*; *Khorozyan et al., 2015*) where livestock are only an alternative prey. However, a number of recent studies have revealed the ability of some large carnivore species to persist in human-dominated landscapes (*Bhatia et al., 2013*; *Ripple et al., 2014*; *Kshettry, Vaidyanathan & Athreya, 2017*). In such cases, predators may heavily depend on anthropogenic food resources, such as domestic animals like dogs, livestock and garbage (*Gehrt, Riley & Cypher, 2010*; *Athreya et al., 2016*; *Kshettry, Vaidyanathan & Athreya, 2018*). In India, reproducing populations of some large carnivores species such as wolves *Canis lupus* (*Jhala & Giles, 1991*), Asiatic lions *Panthera leo* (*Banerjee et al., 2013*), striped hyaenas *Hyaena hyaena* (*Singh, Gopalaswamy & Karanth, 2010*) and leopards *Panthera pardus* (*Athreya et al., 2013*), all of which are capable of attacking livestock and humans, inhabit human-dominated landscapes with dense human populations. This increased spatial overlap between large carnivores and people poses a challenge to conserve them in a country with an average of more than 380 people/km$^2$ (Census of India 2011).

Recent research within the field of human-wildlife conflicts has been increasingly incorporating social science perspectives (e.g., *Dickman, 2010*). These new perspectives have tried to differentiate between the negative impacts that wildlife can have on people (i.e., negative interactions between people and wildlife) from true conflicts that people have about wildlife (i.e., negative interactions between different groups of people about the way wildlife is managed) (*Redpath et al., 2013*). The social science perspectives have also underlined how much of the substance of a human-wildlife conflict can be perceptional and may be only loosely related to actual degree of economic or material impact (*Dickman et al., 2014*). The implication is that while there is a need to carefully investigate each of the different aspects of complex human-wildlife interactions, there is also a need to integrate

the different human perceptions when deriving management recommendations. In this paper, economic losses arising due to shared spaces between people and carnivores are termed 'impacts' whereas hidden differences of opinions and agendas between groups of people is termed as conflict (*Redpath et al., 2013*).

Indian wildlife laws prohibit the killing of wild carnivores such as leopards and hyenas (Anon 1972) and the rural communities are largely accepting their presence (*Sekar, 2011*; *Treves & Bruskotter, 2014*). However, there are many examples where local people either illegally kill large carnivores associated with conflicts or put pressure on wildlife management authorities to remove (translocate) animals from their neighbourhood (*Athreya et al., 2011*). In these contexts, it is clear that there is a need for robust knowledge of carnivore ecology and objective assessments of their impacts on local livelihoods in a human-dominated landscape to guide effective conservation efforts. This is especially true when misguided reactions, such as translocation, can have potentially unforeseen negative effects such as increased attacks on humans by the carnivores near the sites of release (*Athreya et al., 2011*). Unfortunately, there is very little such knowledge available, as wildlife research in India has almost exclusively been focused on protected areas (*Ghosal et al., 2013*). In this study, the levels of impact associated with leopards in a largely rural, agricultural landscape with little wild prey, but supporting high densities of domestic animals and humans was quantitatively assessed. Specifically, we asked the following questions: (i) What were the livestock losses that people faced in the area due to leopard depredation? (ii) What factors predicted livestock attacked by leopards? (iii) Is compensation as a procedure evenly spread among the different groups of people who report losses?

## METHODS

### Study area

The study was carried out in Akole taluka (or county) of Ahmednagar district, located in the western part of Maharashtra State, India (Fig. 1). Approximately 80% of the human population in Ahmednagar district is rural and the major crops grown in the area are sugarcane, millet, and vegetables. Annual rainfall is highly seasonal, and can vary from 1,000 to 2,000 mm, although most farmers have access to irrigation from the river flowing through the valley and from percolation wells widely distributed across the landscape. Akole town had a human population of about 20,000 at the time of our study, while the average population density of the Ahmednagar district is 266 people/km$^2$ (Census of India 2011) and average livestock density of Ahmednagar is 162 /km$^2$ (https://ahd.maharashtra.gov.in/sites/default/files/livestock_census_19th_2012.pdf, accessed on September 2019). Apart from the high density of various types of livestock, the density of domestic dogs which are important food resources for leopards (*Athreya et al., 2016*) is also high at 24/km$^2$. Dogs could be feral or owned but even in the latter case, they are usually free roaming in the day time returning to the homes in the night.

The wild carnivores that are resident in the study area are leopard, striped hyaena, golden jackal (*Canis aureus*), Indian fox *(Vulpes bengalensis)* , jungle cat *(Felis chaus)* and

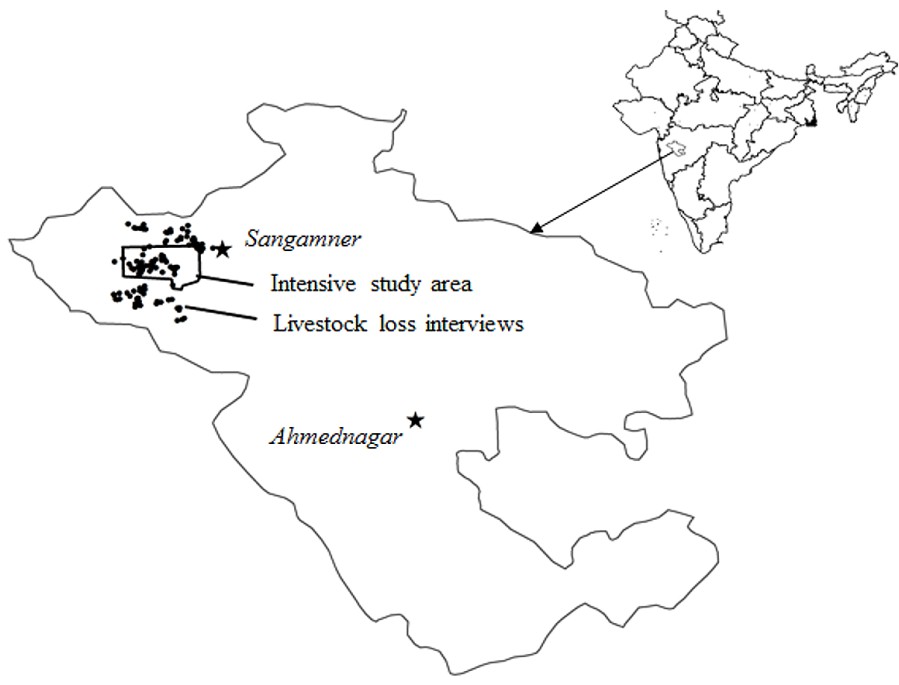

**Figure 1 The Ahmednagar district is shown with the study area marked as a polygon where information on livestock losses was collected from households chosen randomly.** The black dots represent the households which reported livestock loss and claimed compensation.

rusty spotted cat (*Prionailurus rubiginosus*) (*Athreya et al., 2013*) and no wild ungulate species occur. The density of leopards and hyaenas is recorded at five adults of each species per 100 km² (*Athreya et al., 2013*). Many groups of people live in this landscape, the numerically dominant group in the irrigated valley is of resident farmers who own both land and livestock. Migratory shepherds, who are a pastoralist group, arrive each dry season (September to May) with their large herds of sheep and goats to pasture on the post-harvest crop residues in fields, a practice encouraged by the landowners who benefit from organic manure deposited by the animals.

The study was conducted in a 647 km² area of the wider landscape surrounding the 179 km² intensive study area described in *Athreya et al. (2013)*, *Athreya et al. (2016)* which centered around the town of Akole. The nearest wildlife reserve is the Kalsubai Harishchandragarh Wildlife Sanctuary (299 km²) which is 18 km away from the western edge of the study area (Fig. 1).

Oral consent was obtained from the farmers who were interviewed, the reasons for collecting the information was provided to them and their identities were not recorded. All the necessary permissions to carry out the study were obtained from the Maharashtra Forest Department, the government agency responsible for protection and management of all wildlife on public and private lands in the state.

## Information on livestock losses

Official information on the reported number of leopard encounters with humans leading to injuries and claims filed for livestock attacks caused by leopards were obtained from the Maharashtra Forest Department. Based on this information, three different groups of people were interviewed to obtain different kinds of information.

(i) Residents of randomly chosen households "**random households**" within the intensive study area were interviewed to assess the level of under-reporting of livestock losses. They were chosen by dividing the entire study area into grids where all houses were marked and numbered using Google Earth and one house was chosen from each grid using a random number generator. We enquired about the number of domestic animals they owned and if the farmers had applied for livestock compensation in the last five years. Information on livestock losses over five years was obtained because the number of losses in one year was too few to provide meaningful information. If they had not applied for compensation, they were asked to list the reasons.

(ii) People who had lost livestock recently (in the three years between 2006 and 2009) and had filed for compensation and henceforth termed as "**compensation claimants**" were interviewed to assess the extent of their losses and the circumstances in which their livestock was lost to leopards. The farmers in this group were spread over a larger landscape that surrounded the intensive study area. A wider area was sampled in order to provide access to a large enough sample for statistical analysis. Information such as the species, number of domestic animals owned and lost in one year, details of the attack such as location (in cattle sheds, in the open, or while grazing), and location of the feeding area of the animal at night (stall fed or free ranging) were obtained during the interview. The quality of livestock sheds was assessed by visiting the shed where the livestock were kept at night. If the sheds had an opening through which leopards could get inside, or if they were fragile (made of sticks which a leopard could push aside) then the quality of protection was noted as vulnerable and if the shed was completely sealed it was noted as predator-resistant.

(iii) "**Migratory shepherds**" who use the valley in the dry season and are accompanied by extremely large herds of sheep and goats face livestock losses to carnivores (*Agarwala et al., 2010*). Information on their domestic animal holdings, the total livestock mortality in the past year and all the reasons for mortality were requested.

All interviews were semi-structured and were conducted between September 2007 and September 2009. A local farmer who was part of the team was present during the interviews to increase acceptance towards the research team among the locals.

## Analysis

We used Generalized Linear models with a binomial error structure and our inferences were based on model selection using an information theoretic approach (*Burnham & Anderson, 2002*; *Johnson & Omland, 2004*) to examine the following questions;

(i) We assessed the risk of small stock (almost always goats) versus large stock (calf) being attacked by leopards. Each compensation claimant household that faced a livestock attack was considered as a data point, the proportion of the species attacked per household was considered as the response variable and the species of livestock (goat or calf) was

 

considered as the predictor variable. These two groups of animals were chosen because initial analysis indicated that they were mainly predated upon by leopards compared to other species.

(ii) We wanted to assess what variables influenced whether a household would face a livestock predation event by leopards. A predation attempt was considered as the response variable (with a binomial error structure) and the independent covariates were whether the household owned a dog, the number of goats, the number of other livestock apart from goats, and the quality of livestock protection. In this analysis, data from both the "compensation claimants" and the "random households" was used.

All the covariates that were included in the competing models were based on a priori knowledge and hypotheses. The best model was chosen based on the Akaike Information Criterion (AIC) and the model with the smallest AIC was regarded as the best approximating model. In the analyses, no single best model ($\Delta$AIC >10 from the next model) was identified. Therefore, inferences were based on multiple models. In such cases Akaike weights (the weight of a given model relative to other models in the candidate set of models) were calculated for all models to obtain the model averaged parameter estimates and confidence intervals (*Burnham & Anderson, 2002*). The statistical software R 2010 (version 2.11.0; *R Core Team, 2010*) was used for all the analyses.

## RESULTS

### Extent of livestock losses

A total of 337 compensation entries of livestock depredation by leopards was reported to the Forest Department between April 2006 and February 2009 over an area of 647 km$^2$ (Fig. 2). The total number of livestock reported as killed by leopards as per the compensation records within the 179 km$^2$ intensive study area over a three-year period was 242 livestock (165 goats, 61 calves, 13 sheep and 3 adult cows from a total of 224 households). This implies an average of 81 livestock killed each year in a landscape (or 45 per year per 100 km$^2$ ) with a leopard and hyaena density of 10 adults per 100 km$^2$ and a very high density of domestic animals (16,200 heads per 100 km$^2$). The maximum number of livestock killed during one incident was 4 (reported from three households).

Despite the high density of humans and the relatively high density of leopards, no human had been killed by leopards during living memory and accidental attacks resulting in injury were rare in the intensive study area. During the study period (2007–2009), a boy was injured while he was cycling on a path flanked by high grass in which a leopard was sitting. The leopard clawed the boy's leg but did not follow the boy even though the boy was alone. In addition, just after our field work ended, in April 2009, a couple was knocked off their motorbike by a leopard that was trying to cross the road at the same time their bike drove past. Two more men were injured in October 2009 when they were driving their motorbike at night. In this case, two leopards jumped on the motorbike, bit the leg of one person and ran off into the sugarcane fields. However, it was learnt later that this attack followed a provocation where another person had chased one of the leopards (both were GPS-collared by us) in a jeep for nearly 500 m.

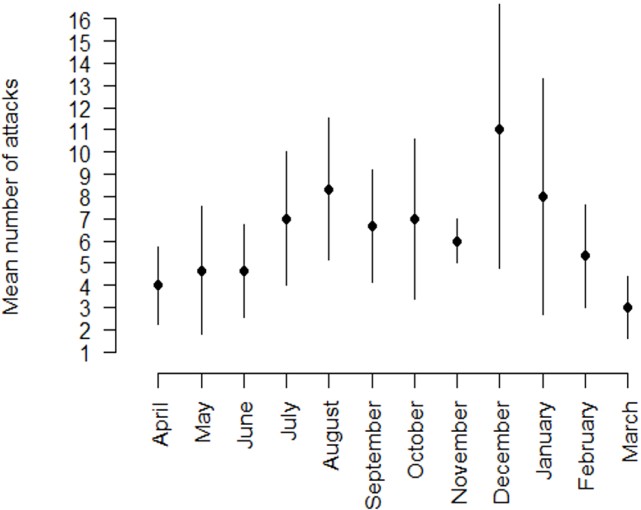

Figure 2 **Seasonal distribution of the average number of livestock (with SD bars) reported as preyed upon by leopards in the intensive study area (179 km²) between April 2006 and February 2009.** The information is from Forest Department records.

Table 1 **Comparison of livestock mortality between random households and compensation claimants.** The random households provided information over five years prior to the interview as their yearly losses were too few to enumerate. The compensation claimants on the other hand provided information for the year prior to the interview.

|  | Random households | | Compensation claimants | |
| --- | --- | --- | --- | --- |
|  | Owned | Leopard kills | Owned | Leopard kills |
| Goats | 224 | 3 | 480 | 107 |
| Sheep | 5 | 0 | 0 | 2 |
| Calves | 70 | 0 | 83 | 17 |
| Adult cattle | 209 | 0 | 293 | 1 |
| Buffalo | 9 | 0 | 13 | 0 |
| Total | 517 | 3 | 869 | 127 |

The results indicate that both sets of farmers, 'compensation claimants' and 'random households', owned similar numbers of livestock ($t$-test, $p = 0.145$) (Table 1). The average number of goats were 2.9 per household (range = 0–40) and calves were 0.9 per household (range = 0–8) whereas buffaloes and sheep occurred in smaller numbers (*Athreya et al., 2016*).

In the five-year period prior to the interviews, the 77 random households that we obtained information from reported losing 1% of their livestock holdings (from a total of 517) to leopards and 9% to other causes of mortality. Their losses to leopard depredation were few; no buffalos, no adult cows, no calves nor sheep were killed by leopards in five years with only three individuals from a total stock of 224 goats killed by leopards in the five-year period. The most common causes of livestock mortality of the "random
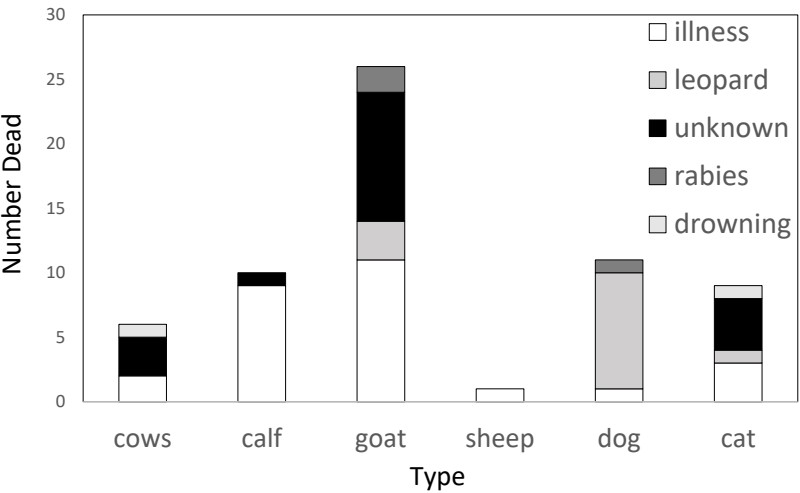

**Figure 3 Causes of livestock mortality among the randomly interviewed farmers (n = 77) from the intensive study area of Akole over a five year period.**

households" were illness and natural causes of death (Fig. 3). The only exception was for domestic dogs, where leopard attacks were by far the dominant cause of death (Fig. 3).

In the case of the "compensation claimants" in the wider landscape, 134 of the households we surveyed owned a total of 869 livestock. They reported 22% of a total of 480 goats and 20% of a total of 80 calves dying due to leopard predation in one year. Among the compensation claimants who lost livestock due to leopard attacks, the probability of a goat being attacked was twice that of a calf being attacked (GLM; probability of a goat being attacked: mean = 0.364; 95% CI [0.148–0.650]; probability of a calf being attacked: mean = 0.176; 95% CI [0.106–0.266]). The 31 "migratory shepherds" owned a total of 4,991 sheep and goats of which 813 (16%) had died in the year prior to the interview. Leopards accounted for 12% of the 813 livestock deaths, the remaining occurring due to natural causes.

## Patterns of livestock losses

From the random households, 75% of the farmers stall-fed their livestock, 18% took them out to graze, and 7% did both (N = 68 provided responses to this question). Seventy-six percent of the random households (N = 68 provided responses) protected their smaller livestock in leopard-resistant sheds or had people sleeping near the livestock at night (a traditional method of protecting their livestock), while the remaining 24% of the respondents did not have leopard-proof sheds. Dogs are usually not protected and are almost always left loose outside the houses in the night, although this was not quantified.

Information from the compensation claimants indicates that most leopard attacks on livestock took place during the night (83%; 103 farmers responded to this question), and all nighttime attacks occurred in the residential compound of the farmers (Table 2). The proximity of leopard attacks to the house also allowed the farmers to intervene and in 55% (of the 94 farmers who responded to this question) of the cases the domestic animal was

**Table 2 Time and location details of livestock attacks (number of attacks) by leopards obtained from the compensation claimants.** The total number of responses we got for this question was from 103 claimants.

|  | Day time (0600–1800) | Night time (1800–0600) | Do not know | Total |
|---|---|---|---|---|
| Grazing | 12 | 0 | 0 | 12 |
| Residential area | 1 | 78 | 0 | 79 |
| Do not know | 1 | 4 | 0 | 5 |
| Total | 14 | 82 | 7 | 103 |

retrieved. This was done for two reasons, (i) in an attempt to save the animal's life, or (ii) if it was dead, they required the carcass as evidence to file for compensation from the Forest Department. In none of these cases was the farmer attacked by the leopard which is likely to have been in the vicinity.

Our analysis (Table 3) indicates that the important predictors of a livestock predation by leopards are the presence of dogs and the quality of livestock protection, that is, livestock attacks by leopards were more likely if dogs were present in the household and if the livestock protection enclosure was not predator-resistant.

Among the people who had predator-resistant sheds and yet lost livestock, it was due to reasons such as delaying taking the goats into the shed in the evening or failing to put the livestock inside the shed for that night. Furthermore, the analysis also shows that although the average number of goats was higher (5.05 goats; range = 0 to 65) in houses that were visited by leopards to take livestock than in houses (2.9 goats; range = 0 to 40) which did not face livestock damage due to leopards , model averaged coefficients (Table 4) provide poor evidence for an effect of the number of goats on the probability of livestock predation by leopards.

## Compensation payment for livestock losses

In the past, eight of the 77 random households lost their livestock to leopards of which five households did not apply for compensation. The reasons why they did not apply for compensation included, (1) they did not know how to file the complaint, (2) compensation was rarely provided in time, (3) the amount paid was too small and many visits had to be made to the forest officer before it was given, (4) they did not have time to undertake the procedure, and (5) they had reported the loss to the local forest guard but he did not take action.

Sixty eight percent of the migratory shepherds lost livestock (of the 31 interviewed) to leopards in the year before the interview of which 76% of the 21 shepherds (who replied to the question) did not apply for compensation. Half of the shepherds who had their livestock predated upon by leopards did not know they could apply for compensation (10 of 21), four shepherds said they did not have the time to file the cases, and four said that the procedure was too drawn-out and cumbersome. Three mentioned that filing for compensation was too expensive and two said they did not get it when they had filed for it.

**Table 3  Generalised Linear Models (binomial errors) where the response variable that was modelled was the probability of a leopard attack on livestock.** The data set consists of both compensation claimants as well as random households.

| Model | Deviance | AICc | ΔAICc | w |
|---|---|---|---|---|
| Dogs present+livestock protected+dogs present: livestock protected | 119.76 | 128.03 | 0.00 | 0.27 |
| Dogs present + livestock protected | 123.04 | 129.21 | 1.17 | 0.15 |
| Dogs present + number of goats + livestock protected + dogs present: livestock protected | 119.01 | 129.43 | 1.39 | 0.14 |
| Dogs present + number of other livestock + number of goats + livestock protected + dogs present: livestock protected | 116.96 | 129.55 | 1.52 | 0.13 |
| Dogs present + number of other livestock + livestock protected | 121.88 | 130.16 | 2.12 | 0.09 |
| Dogs present + number of goats + livestock protected | 122.07 | 130.35 | 2.31 | 0.09 |
| Dogs present + number of other livestock + number of goats + livestock protected | 120.64 | 131.06 | 3.03 | 0.06 |
| Livestock protected | 128.06 | 132.14 | 4.11 | 0.03 |
| Number of goats + livestock protected | 127.23 | 133.40 | 5.37 | 0.02 |
| Number of other livestock + livestock protected | 128.03 | 134.19 | 6.16 | 0.01 |
| Number of other livestock + number of goats + livestock protected | 127.19 | 135.47 | 7.43 | 0.01 |
| Dogs present + number of goats | 191.08 | 197.25 | 69.21 | 0.00 |
| Dogs present + number of other livestock + number of goats | 190.84 | 199.12 | 71.08 | 0.00 |
| Dogs present | 195.85 | 199.93 | 71.90 | 0.00 |
| Dogs present + number of other livestock | 195.73 | 201.89 | 73.86 | 0.00 |
| Number of goats | 199.29 | 203.37 | 75.33 | 0.00 |
| Number of goats + number of other livestock | 198.98 | 205.15 | 77.11 | 0.00 |
| Number of other livestock | 203.15 | 207.24 | 79.20 | 0.00 |

**Table 4  Protection related factors where the response variable modelled was the probability of a leopard attack on livestock assessed using Generalised Linear Models with binomial errors.** The reported figures are: Model-averaged estimate and 95% confidence interval (CI) of parameters representing factors affecting the chances of leopard attacks on livestock held in farmer's pens. Interview data from both, 'compensation claimants' and 'random households' were used. The interviews were held between September 2007 and September 2009.

| Model averaged parameters | Coefficient | Lower CI | Upper CI |
|---|---|---|---|
| Dogs present | 1.93 | 0.0771 | 3.79 |
| Number of livestock except goats | −0.0844 | −0.229 | 0.06 |
| Number of goats | 0.101 | −0.107 | 0.309 |
| Protection not effective | 3.85 | 2.48 | 5.22 |
| Dogs present: protection not effective | −1.93 | −4.01 | 0.16 |

# DISCUSSION

Previous ecological studies from this area have reported novel findings of a relatively high density of leopards sharing the same space with a high density of humans (*Athreya et al.,*

*2013*; *Athreya et al., 2016*) as well as the complete dependency of leopards on domestic animals (livestock and dogs contributing as much as 87% of the prey biomass) as prey (*Athreya et al., 2016*). Results from the current work indicate that the intensive study area is not unique in hosting a leopard population because a much larger area of nearly 650 km² also reported leopard depredation on livestock, confirming their presence across a much wider landscape. Across India, leopards are known to be responsible for a large number of encounters with humans leading to casualties (*Kshettry, Vaidyanathan & Athreya, 2017*; *Packer et al., 2019*). However, at this study site, only three human injuries caused by leopards were reported during the three-year study period. In two cases they were accidental incidents and the third was a provoked response. In all cases, the animals did not kill the people although it would have been easy for them to do so since there were no other humans in the vicinity at the time of the incidents. No respondent could recall any incident of predation on humans by leopards at the study site. This is despite close interactions between humans and leopards with all the night time predation on livestock occurring close to houses and half of the predated livestock being retrieved by the farmers by chasing the leopards off their livestock which had been attacked. *Odden et al. (2014)* also document using GPS telemetry how close the leopards were to humans both by night and day, implying that there was a constant potential for aggressive encounters between leopards and people.

One would expect livestock losses to be very high because of the absence of wild herbivores in this landscape (*Kshettry, Vaidyanathan & Athreya, 2018*). However, 242 livestock killed in the 179 km² intensive study area were reported as killed by leopards over three years despite extremely high densities of livestock (*Athreya et al., 2016*), and a high density of leopards (*Athreya et al., 2013*). For the average density of 60 houses per km² this implied a loss of 0.45 livestock per km² per year. The interview data also found that most of the farmers protected their livestock effectively which is known to be a very important factor in reducing livestock losses (*Lichtenfeld, Trout & Kisimir, 2015*; *Manoa & Mwaura, 2016*). Therefore, the high density of livestock might not directly lead to high rates of livestock depredation. It clearly also depends on other available food sources (domestic dogs in this case) and how accessible the livestock is to the leopards (*Athreya et al., 2016*). *Frank, Woodroffe & Ogada (2005)* found that traditional daytime and night-time husbandry practices in Kenya are most effective at reducing livestock losses to large predators, findings supported by many other studies (*Meena et al., 2011*; *Hazzah et al., 2014*).

An interesting finding from our study was that the presence of dogs around farms increased the risks of leopard predation on livestock. A common finding in other studies has been that the presence of dogs, especially livestock guarding dogs, can deter large predator attacks (*Gehring et al., 2010*; *Miller et al., 2016*). Dogs trained to keep away predators were also used by the migratory shepherds in our study area (V Athreya, 2008, pers. comm.) but these were largely to keep away wolves and not leopards. However, the dogs used by the resident farmers were generally smaller and not bred as guarding dogs and kept mostly as companions/pets. This small size, coupled with dogs being an important component of leopards' diet in this landscape (*Athreya et al., 2016*) implies that not only are they not a

deterrent, but are in fact an attractant. Indirectly, the presence of so many vulnerable dogs is probably one of the factors that may have led to the persistence of such a dense leopard population in the study area.

In the randomly sampled households the probability of experiencing a livestock loss to leopards was seen to be small but among the compensation claimants, the proportion of livestock lost to depredation was relatively large. As with many other studies, our results also indicate that livestock loss to predators are a small fraction of the total losses farmers face. A review in 2010 of 18 studies of large cat predation found that usually no more than 5% of the total livestock holding was lost to large cat depredation although in a few cases 12% loss was reported (*Loveridge et al., 2010*). *Loveridge et al. (2010)* reports that livestock loss from diseases and theft are usually much higher than to predation by large felids. The random households in our study lost one percent of their livestock over five years to leopard predation and nine percent to diseases and natural causes. In the case of migratory shepherds, 12% of their livestock losses were due to predation in one year but the remaining 88% was lost to other causes. Although individual losses are likely to be severe for marginal farmers, conservation studies dealing with conservation conflicts need to take into account losses of livestock due to all reasons, including disease etc., as a measure of the impact of focal carnivore species on overall livestock losses.

This does not take away from the potential impact of leopard depredation for individual, marginal farmers. As a result of these impacts India, like many countries, has created a compensation system which is designed to redistribute the costs of protected species from the individual farmer to wider society (*Gebresenbet et al., 2018*). As well as representing a form of distributive justice, the philosophy (albeit rarely demonstrated) is that compensation will reduce the antipathy of the affected farmers towards the carnivore (*Dhungana et al., 2016*). However, compensation can also lead to slack herding measures because the farmer expects to get compensated in case of loss (*Bulte & Rondeau, 2007*). For example, compensation payments were stopped in Kenya because of poor financial controls and corruption (*Hazzah et al., 2014*), which is also often a problem in India. Few of these schemes are ever monitored to assess their efficacy (*Nyhus et al., 2005*).

Our results indicate that while under-reporting of loss is very low among the resident farmers (eight houses in 77 did not report losses over a period of five years), it is very high among the migratory shepherds (68% of 31 interviewed reported leopard predation incidents in one year) and 76% of those that lost livestock to leopards did not apply for compensation. The fact that some respondents in our study area cited the procedure as cumbersome, resulting in low payment for too much effort, and did not bother to apply for compensation is an indication of procedural problems with the current system, especially in the case of the migratory shepherds. Large amounts of government funds are spent on compensation each year by the state governments without monitoring the delivery system or the effect of its intention (to make people more tolerant to losses). Recently Maharashtra has improved the delivery system where previous cash payments are now provided as electronic transfers directly into the farmers' accounts.

However, although it can be argued that gains could be made through adjustments to the procedures associated with compensation payments, it is also important to consider whether

greater gains could be made by moving away from paying for losses, and instead move to a system of paying for, and assisting with prevention of livestock losses. Compensation rates will always increase due to increasing costs of livestock but if the same resources are used for proactive measures that aim at preventing a loss, it is likely to result in improved longer term mitigation as well as reduce the drain on funds and put the onus of protection on the farmer rather than the state agency that represents the wild animal. More importantly, it will reduce losses to individual farmers, many of whom are marginal and for whom livestock is an integral part of their livelihood generation (*Agarwala et al., 2010*). Existent livestock protection practices in the landscape implies that only minor modifications are required to make them effective.

## CONCLUSION

Carnivores are often viewed as dangerous and incompatible with human-dominated landscapes. It has also been argued that there are some large cat individuals who are 'problem' animals and their removal will ease livestock depredations (*Linnell, 2011*). By this definition all the leopards in our study site would be categorised as 'problem' animals because they all depend at least partly on domestic animals for prey due to the absence of any suitable wild prey. However, our results show that they are residing in the area with an impact that is unexpectedly low considering the density of humans, their livestock and leopards. Should these leopards still be termed as problem animals? The term conflict is most commonly used in conservation literature while describing damage incidents of livestock to predators, which are often less compared to losses due to disease or illness. Conflict implies that the predator is implicated as an actor who is at "fault" whereas in reality the livestock loss has occurred because of ineffective protection of livestock by humans. It is important that we shift the onus of responsibility to the owners of the livestock rather than on the predator who will attack any herbivore/carnivore without distinguishing whether it is wild or domestic if it is available. Recently several authors (*Redpath et al., 2013*; *Davidar, 2018*) have called for the word "conflict" to be only used for the human-human aspect of conservation conflicts, preferring to reserve the word "impact" for the material and economic effects of predators on people. In our study site there were relatively few social conflicts linked to the leopards (*Ghosal & Kjosavik, 2015*), and impacts were relatively low and widely dispersed across the human population. In many ways, the discourse could be even more constructively switched away from impacts to one of a failure to completely adapt to the presence of leopards (*Carter & Linnell, 2016*). However, when compared to many other case studies of human-large felid interactions and the associated conflicts, the situation in Akole is very close to coexistence (sensu *Carter & Linnell, 2016*), and the required adjustments to human practices are minor. Now is certainly the time to make the necessary adjustments, as it is likely to be far easier to prevent escalating impacts, and potential conflicts from developing than reverse them once they have appeared (*Miller et al., 2016*).

# ACKNOWLEDGEMENTS

We would like to thank the Maharashtra Forest Department for their collaborative support to this project. We would like to especially thank Mr B Majumdar (retired Chief Wildlife Warden), Mr P Thosre (Retired Additional PCCF), Mr Saiprakash (PCCF) and Mr Phatangare (DFO Wildlife Pune). The work was only possible, and even more enjoyable, due to the presence of Mr Ghule who was part of our field team.

## Funding

The study was funded by a Kaplan Graduate Award (Panthera), the Wildlife Conservation Society, the Royal Norwegian Embassy in New Delhi, the Norwegian Institute of Nature Research, the Research Council of Norway (grants 201693 and 251112) and the Asian Nature Conservation Foundation. The funders had no role in study design, data collection and analysis, decision to publish, or preparation of the manuscript.

## Grant Disclosures

The following grant information was disclosed by the authors:
Kaplan Graduate Award (Panthera).
The Wildlife Conservation Society.
The Royal Norwegian Embassy in New Delhi.
Norwegian Institute of Nature Research.
The Research Council of Norway: 201693, 251112.
Asian Nature Conservation Foundation.

## Competing Interests

The authors declare there are no competing interests.

## Author Contributions

- Vidya Athreya conceived and designed the experiments, performed the experiments, analyzed the data, prepared figures and/or tables, authored or reviewed drafts of the paper, and approved the final draft.
- Kavita Isvaran analyzed the data, prepared figures and/or tables, and approved the final draft.
- Morten Odden conceived and designed the experiments, analyzed the data, prepared figures and/or tables, authored or reviewed drafts of the paper, and approved the final draft.
- John D.C. Linnell conceived and designed the experiments, authored or reviewed drafts of the paper, and approved the final draft.
- Aritra Kshettry analyzed the data, authored or reviewed drafts of the paper, addressed reviewers comments from previous submission, and approved the final draft.
- Jagdish Krishnaswamy conceived and designed the experiments, authored or reviewed drafts of the paper, critical comments on manuscript, and approved the final draft.

- Ullas K. Karanth conceived and designed the experiments, authored or reviewed drafts of the paper, and approved the final draft.

## Human Ethics

The following information was supplied relating to ethical approvals (i.e., approving body and any reference numbers):

No approval was obtained since no sensitive information was sought from the subjects. However, the questionnaire was approved by the Doctoral Advisory Committee of Dr. Vidya Athreya as this study was part of her PhD.

## Data Availability

The raw data is available in the Supplementary File.

## Supplemental Information

Supplemental information for this article can be found online at http://dx.doi.org/10.7717/peerj.8405#supplemental-information.

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
