# Peer review of "The impact of leopards (Panthera pardus) on livestock losses and human injuries in a human-use landscape in Maharashtra, India"

_PeerJ, doi:10.7717/peerj.8405_

## Round 0.1 · original submission · Major Revisions

I received comments of two reviewers. Both of them think your manuscript needs a major revision to be accepted. The reviewers made several excellent suggestions on how to improve the description of key terms, the flow of your arguments and the quality of your analyses. If you decide to revise the work, please submit a list of changes or a rebuttal against each point which is being raised by both reviewers when you submit the revised manuscript. You should expect a second round of revision for your manuscript.

·

Basic reporting

No comment

Experimental design

No comment

Validity of the findings

No comment

Additional comments

The authors present an interesting manuscript about the relationships between leopards and humans in the agricultural landscapes of Western Maharashtra in India. The article is well written, in clear and professional English, the context is clearly provided with a good amount of references. I found the article to be well structured and the authors have shared their raw data. The results are relevant to the hypotheses that are clearly stated at the beginning of the manuscript. My major concern is that I find that the authors do not completely answer the question posed in the title of their paper: “When does damage become conflict?”. Some parts of the discussion could be removed to leave more space to better answer this question.

I also have a few minor concerns that I would like to see addressed before recommending this manuscript for publication.

Line-specific comments

Line 2: in the title, add “India” after Western Maharashtra.
Line 30: human-leopard relationships.
Line 31: Please, add the average livestock density.
Line 37: What kind of dogs? Surely not guard dogs? Is it because leopards consider them as prey species?
Lines 41-43: Please be a bit more specific as these recommendations are very broad.
Line 70: The Gehrt et al. (2010) reference is about mesocarnivores, not large carnivores. Please either find another reference of remove “large” in line 69.
Line 72: Do you consider that domestic animals are pets (dogs, cats) as well as livestock? It might be worth defining what the authors mean by “domestic animals”.
Lines 88-91: Please rephrase as it is not very clear what you are trying to say.
Line 98: “(…) for carnivore ecology, notably their diet, and objective (…)”
Line 101: Negative effects on what? Please, state it clearly here.
Line 109: ? instead of . after “who report losses”.
Line 109-111: Please check the grammar in this sentence.
Line 118: Add information about the climate. Is it tropical?
Line 120-122: Clearly state how the human density of your study site relates to that of the district (you state it is 400 inhabitants/km2 in the abstract).
Line 125: Spacing between Felis and chaus.
Line 145: Official information on the number …
Lines 145-146: Are the claims for leopard predation only or for any types of predation by any type of carnivores (large and medium-sized)? Please add this information.
Line 173: Do these shepherds keep the livestock of several owners or is it their own livestock?
Line 179-180: Please explain why it was important to have this farmer with you during the interviews. Is it because it increased trust?
Line 197: Here again, please be clearer about “dog”. Do you mean dogs kept as pets?
Line 214: “between April 2006 and February 2009”.
Line 215 and 218: Are the compensation records for leopards and hyaenas or leopards only?
Line 218: 10 adults per 100 km2.
Line 234: Add a statistical test allowing you to say that.
Lines 255-256: Is there no deaths caused by mesopredators in your study area?
Line 307: Remove “very”
Line 306: Add “Previous” before “Ecological studies”.
Lines 308-311: Is it really the most important aspect of your manuscript? The following sentences seem a lot more important, along with the fact that despite no wild prey and a high density of livestock, leopards do not seem to attack livestock very often.
Line 311: Check spacing between words.
Line 327: Please, give the density of livestock somewhere in the manuscript, preferentially in the Description of the study area.
Line 328-329: It would be good to put that in perspective with other published data on leopard predation on livestock in other places, maybe East Africa?
Line 333: Rephrase “automatic cause”.
Line 335: Remove “G laurence” to the citation.
Line 366-376: I feel that this paragraph is not really necessary in your manuscript as you are not really presenting any results directly linked to compensation in your study. I would thus remove this paragraph.
Line 391: Replace if by whether.
Lines 399-401: Rephrase, “The fact that most livestock are already enclosed at night implies that only minor enclosure improvements will be needed to exclude leopards”.
Lines 404-405: Please rephrase as the sentence is not very clear.
Line 406: Landscapes.
Line 429: Check spacing of words.

Table 1: Please give the years that the information was provided for.
Table 2 can be in supplementary material if not enough space in the paper itself.
I could not review Figure 1 and Figure 3 because of their very low definition. Make sure a higher definition is used next time.

Reviewer 2 ·

Basic reporting

The manuscript was written well. Please see below comments on a few figures / tables not referred to the manuscript and areas where references are missing.

Experimental design

Methods were completed very well and aims are clearly stated.

Validity of the findings

The manuscript was very well written and presented some interesting results. Well done! The finding that domestic dogs increase the likelihood of a leopard attack was particularly fascinating. It was very good how you included suggested conservation actions, especially in relation to mitigating other causes of loss such as disease and discuss avoiding placing intentional blame on leopards for attacks. My main critique was the framing of the manuscript within the definitions of damage / conflict. Although I liked the idea behind this approach, I felt that the terms were poorly laid out and confusingly used throughout. I also felt that you did not properly investigate these terms with the respondents. What were the farmers and shepherds’ views on what constituted conflict / damage or a problem animal? What was their threshold of when does damage become conflict? This was what I expected to find out when the aims were stated but this was not included. Unless this is more adequately addressed, I suggest that the premise of exploring at what point negative wildlife impacts on people actually constitute conflict is dropped as this was not supported well enough with your current set of results. Although the manuscript would loose an interesting angle, it would still stand up as well executed interesting science without this additional dimension.

Additional comments

Please see specific points to address below:


Line 37 – the word ‘probably’ should be changed to probability

Line 83 / 84 – the sentence about the difference between damage and conflict is not well grounded and therefore seems confusing after you have been using the term conflict to imply damage caused by carnivores in previous paragraphs. In lines 418 -421 of the discussion you explain the difference in the literature between damage / impact and conflict better. It would be beneficial to integrate this sentence from the discussion into the introduction to provide the necessary explanation of the terms earlier on. It is a bit confusing how you don’t seem to consistently differentiate between damage and conflict in the manuscript since this is the premise of the argument. It is probably worth swapping the order of the introduction around to open with a discussion of damage / conflict and very clearly lay out your terminology and definitions from the onset so you adhere to them throughout.

Line 93 – remove the words ‘the attitudes of’. This is implied in the sentence.

Line 142 – move the description of the Maharashtra Forest Department included in lines 146 – 147 to the first time the organisation is mentioned in line 142.

Line 145 – add in the word ‘reported’ before ‘leopard encounters’

Line 174 – the statement that migratory shepherds face livestock losses to leopards either needs a reference or it is a finding that is only revealed in the results section and should be omitted here.

Line 187 – Please turn this first line into a proper sentence.

Line 217 – put the word ‘adult’ in front of the word ‘cow’ throughout the manuscript to differentiate between adult cow and calf.

Line 245 – You list Figure 3 here but I did not see Figure 2 referred to in the manuscript prior?

Line 269 - You list Table 2 here but I did not see Table 1 referred to in the manuscript prior?

Line 277 – Should Table 2 listed here be Table 1?

Line 308 – Please include a reference to support the statement that leopards have a complete dependency on domestic animals as prey. It would be interesting to state how much of their diet is comprised of domestic dogs compared to other livestock.

Line 335 – the reference Frank, Woodroffe and Ogada, 2005 should not include G laurence.

Line 341 – Please include references to support the sentence that other studies found that the presence of dogs can deter large predator attacks.

Line 344 – Why are these domestic dogs kept? As companion animals or for another purpose? If you suggest that farmers stop keeping dogs to reduce leopard attacks, do you think that a reduction in dogs would lead to an increase in livestock consumed instead?

Line 361 – 364 – This sentence about mitigating other causes of losses like disease as a means to reduce conflict with carnivores needs a little more explanation to clarify it. Including a bit about how farmers may become more acceptant about a small percentage of livestock lost to predators if they didn’t loose so many due to other causes would be helpful to explain this better. If there is a reference to support that hypothesis that would be even better.

Line 378 – The migratory shepherds are extremely interesting and it would be good to include a bit more in the discussion about why their losses were so much higher than the resident farmers and perhaps differentiating between conflict and damage for shepherds as opposed to the other groups.

Line 393 – It would be worth talking about how to improve conflict prevention for migratory shepherds since they are less able to have predator resistant enclosures and the presence of dogs may not be an attractant for leopards in the same way.

Line 410 – Why are leopards only creating minimal economic impact? Any ideas?

Line 411 – The question should these leopards be termed as problem animals seems very subjective and may be answered very differently by members of different groups or people with different backgrounds / experiences. Did you ask any of the interviewees whether they considered the leopards to be problem animals? Were you able to ascertain what the threshold for this definition might generally be? This is important to meet your currently stated aims (Line 109 - 111).

Line 423 – You state that the impacts were relatively low and widely dispersed. However, 68% of migratory shepherds lost livestock to leopards. This seems quite high and not very evenly dispersed across the groups. To these people the impact of these losses might have big implications on their livelihoods and income. Even a farmer who looses only one animal to predation might feel that the impact is large if they can’t afford that loss. A loss of animal might have other implications as well aside from lost income. It can also represent a lost sense of security, lost time, and emotional strain. I think that it seems quite subjective to call the losses low when you haven’t ascertained what the losses really mean to the interviewees.

Line 426 - 427 – You state that the situation is close to coexistence but again the migratory shepherds and compensation claimants had high levels of livestock losses to leopards. This does not seem like only minor adjustments to human practices are required for them. Are there any retaliatory measures taken against leopards at the study site as a result? This should be mentioned in the manuscript.

Line 429 – You state is easier to prevent conflicts developing. Again it is confusing which definition of conflict you are using. Do you mean human-human conflict or impacts between humans and leopards? This needs to be refined throughout.

Table 1 - Please fill in the number of leopard kills of buffalo by compensation claimants. As previously mentioned, I did not see Table 1 mentioned in the manuscript.

Table 2 – Please fill in the blank spaces in the Do not know and Total columns. Mention in the heading why you only included data from 103 attacks.

Table 3 – Why was ‘dogs present’ listed twice in the top ranking model and the 4th ranking model? Make sure all models listed are consistently capitalised at the beginning.

Figure 2 - As previously mentioned, I did not see Figure 2 mentioned in the manuscript. The findings presented about seasonal differences should be explored in the results and discussion section if you would like to retain this figure.

---

## Round 0.2 · Minor Revisions

There are still a few minor corrections that you need to do before it is accepted for publication. Please, follow all the suggestions made by the reviewer.

·

Basic reporting

The authors have answered all my questions and have made the changes that I suggested. I am therefore happy to recommend this article for publication in PeerJ once these last minor changes have been taken into account:

1. I just want to make sure that the new title is "The impact of leopards (Panthera pardus) on livestock loss and human injuries in a human use
landscape in Maharashtra, India (#38063)" as there were two different titles in the pdf and word versions. I would use "losses" rather than "loss" and please check if you should write "human-use" rather than "human use".

2. Well done for making recommendations. In the abstract, l.39, you recommend "informing the public". After reading your paper, I would rather recommend informing the migratory shepherds because as you said, there are no human deaths due to leopards in your study area. However, the possibility of being compensated does not seem to be known by everyone (neither the way to do it). Another important recommendation would be to focus on the other causes of livestock deaths. Maybe some of them could be easily tackled (with vaccinations for example).

3. l. 68 "like dogs, livestock and garbage"

4. l. 91, "these carnivores": it is not obvious what species you are referring to. Is it the species you list at l. 69-72.

5. l. 105 "due to leopard depredation"

6. l. 106, please rephrase: (ii) what factors predisposed a farmer to leopard attacks on his livestock.

7. l.120 /km2 instead of per sq km

8. l. 124: if you say that it is high, you should at least give an approximate number. If there is no number, rather say "is thought to be high but no estimation exists to date".

9. l. 149 livestock losses

10. l. 153 "kinds of information :"

11. l. 191 "the risk of small stock versus large stock", remove the "a" or put it everywhere.

12. l. 202 remove the coma after "both".

13. l. 170 "at night"

14. l. 334-335 There is a problem with the sentence: reformulate or remove "recorded in any human dominated landscape".

15. l. 352, check spacing

Experimental design

No comment

Validity of the findings

No comment.

Additional comments

Well done on a very interesting and useful paper!

---

## Round 0.3 · accepted · Accept

I checked the new version of your paper, and I think that it is ready for publication. Congratulations!